# FastCorrect: Fast Error Correction with Edit Alignment for Automatic Speech Recognition

**Yichong Leng**[1][*]**, Xu Tan**[2]**, Linchen Zhu**[3]**, Jin Xu**[4]**, Renqian Luo**[1]**, Linquan Liu**[3]
**Tao Qin**[2]**, Xiang-Yang Li**[1]**, Edward Lin**[3]**, Tie-Yan Liu**[2]
[1]University of Science and Technology of China, [2]Microsoft Research Asia
[3]Microsoft Azure Speech,[4]Tsinghua University
[1]{lyc123go,lrq}@mail.ustc.edu.cn,xiangyangli@ustc.edu.cn
[2]{xuta,taoqin,tyliu}@microsoft.com
[3]{linczhu,linqul,edlin}@microsoft.com
[4]j-xu18@mails.tsinghua.edu.cn

## Abstract

Error correction techniques have been used to refine the output sentences from automatic speech recognition (ASR) models and achieve a lower word error rate (WER) than original ASR outputs. Previous works usually use a sequence-to-sequence model to correct an ASR output sentence autoregressively, which causes large latency and cannot be deployed in online ASR services. A straightforward solution to reduce latency, inspired by non-autoregressive (NAR) neural machine translation, is to use an NAR sequence generation model for ASR error correction, which, however, comes at the cost of significantly increased ASR error rate. In this paper, observing distinctive error patterns and correction operations (i.e., insertion, deletion, and substitution) in ASR, we propose FastCorrect, a novel NAR error correction model based on edit alignment. In training, FastCorrect aligns each source token from an ASR output sentence to the target tokens from the corresponding ground-truth sentence based on the edit distance between the source and target sentences, and extracts the number of target tokens corresponding to each source token during edition/correction, which is then used to train a length predictor and to adjust the source tokens to match the length of the target sentence for parallel generation. In inference, the token number predicted by the length predictor is used to adjust the source tokens for target sequence generation. Experiments on the public AISHELL-1 dataset and an internal industrial-scale ASR dataset show the effectiveness of FastCorrect for ASR error correction: 1) it speeds up the inference by 6-9 times and maintains the accuracy (8-14% WER reduction) compared with the autoregressive correction model; and 2) it outperforms the popular NAR models adopted in neural machine translation and text edition by a large margin.

## 1 Introduction

In recent years, error correction techniques [1, 5, 22, 26, 33] have been widely adopted to refine the output sentences from an ASR model for further WER reduction. Error correction, a typical sequence to sequence task, takes the sentence generated by an ASR model as the source sequence and the ground-truth sentence as the target sequence, and aims to correct the errors in the source sequence. Previous works on ASR error correction [22, 26] usually adopt an encoder-decoder based autoregressive generation model. While achieving good WER reduction, autoregressive models suffer from slow inference speed, and do not satisfy the latency requirements for online ASR services. For

---

[*]This work was conducted at Microsoft. Corresponding author: Xu Tan, xuta@microsoft.com

35th Conference on Neural Information Processing Systems (NeurIPS 2021).

example, the latency of our internal product ASR system is about 500ms for an utterance on a single CPU, but the latency of the autoregressive correction model alone is about 660ms, which is even larger than the original ASR system and unaffordable for online deployment.

Non-autoregressive (NAR) models can speed up sequence generation by generating a target sequence in parallel, and attract much research attention, especially in neural machine translation (NMT) [6, 8, 9, 41]. Unfortunately, direct application of NAR models designed for NMT to ASR error correction leads to poor performance. According to our experiments, using a popular NAR model from NMT [9] for error correction even increases WER, i.e., the correction output is worse than the original ASR output. Different from NMT where almost all input tokens need to be modified (i.e., translated to another language), the modifications in ASR correction are much fewer but more difficult. For example, if an ASR model achieves 10% WER, only about 10% input tokens of the correction model need to be modified, and these tokens are usually difficult to identify and correct since they have already been mistaken by the ASR model. Thus, we need to take the characteristics of ASR outputs into consideration and carefully design NAR models for ASR error correction.

In ASR error correction, the source and target tokens are aligned monotonically (unlike shuffle error in neural machine translation), and ASR accuracy is usually measured by WER based on the edit distance. Edit distance provides the edit and alignment information such as insertion, deletion and substitution on the source sentence (the output of an ASR model) in order to match the target (ground-truth) sentence, which can serve as precise guidance for the NAR correction model. Based on these observations, in this paper, we propose FastCorrect, a novel NAR error correction model that leverages and benefits from edit alignment:

- In training, FastCorrect first obtains the operation path (including insertion, deletion and substitution) through which the source sentence can be modified to target sentence by calculating the edit distance, and then extracts the token-level alignment that indicates how many target tokens correspond to each source token after the insertion, deletion and substitution operations (i.e., 0 means deletion, 1 means unchanged or substitution, $\geq 2$ means insertion). The token-level alignments (token numbers corresponding to each source token) are used to train a length predictor and to adjust the source tokens to match the length of the target sentence for parallel generation.

- In inference, we cannot get token alignments as the ground-truth sentence is not available. We use the length predictor to predict the target token number for each source token and use the predicted number to adjust the source tokens, which are then fed to the decoder for target sequence generation. With this precise edit alignment, FastCorrect can correct the ASR errors more effectively, using a length predictor to locate which source token needs to be edited/corrected and how many tokens will be corrected to, and then using a decoder to correct the tokens correspondingly.

Since current ASR models have already achieved high accuracy, there might be not many errors in ASR outputs to train a correction model, even if we have large-scale datasets for ASR model training. To overcome this limitation, we use the crawled text data to construct a pseudo correction dataset by randomly deleting, inserting and substituting words in the text data. When substituting word, we use a homophone dictionary considering that ASR substitution errors are mostly from homophones. Those randomly edited sentences and their original sentences compose the pseudo sentence pairs for correction model training. In this way, we first pre-train FastCorrect on the large-scale pseudo correction dataset and then fine-tune the pre-trained model on the limited ASR correction dataset.

The contributions of this work are as follows:

- To our knowledge, we are the first to propose NAR error correction for ASR, which greatly reduces the inference latency (up to 9×) compared with its autoregressive counterpart while achieving nearly comparable accuracy. Our method also outperforms the popular NAR models adopted in machine translation and text edition by a large margin.

- Inspired by the distinctive error patterns and correction operations (i.e., insertion, deletion and substitution) in ASR, we leverage edit alignments between the output text from ASR models and the ground-truth text to guide the training of NAR error correction, which is critical to FastCorrect.

The code of FastCorrect is available at `https://github.com/microsoft/NeuralSpeech/tree/master/FastCorrect`.

## 2 Background

**Error Correction**   In the field of natural language processing, error correction aims to correct the errors in the generated sentence by another system, such as automatic speech recognition [26, 31, 33, 11], neural machine translation [36] and optical character recognition [27]. The error correction models for ASR can be divided into two categories based on whether the model can be trained in an end-to-end manner. Based on the method of statistic machine translation, Cucu et al. [4] performed error correction for ASR system. D'Haro and Banchs [5] proposed to use a phrase-based machine translation system to serve as a correction model for ASR. Anantaram et al. [1] used a four-step method to repair the ASR model output by ontology learning. With the increasing of training corpus, end-to-end correction models are more accurate and become popular. A language model was trained for ASR correction in Tanaka et al. [39], which could exploit the long-term context and choose better results among different ASR output candidates. Mani et al. [26] utilized a Transformer-based model to train an ASR correction model in an autoregressive manner. Liao et al. [22] further incorporated the MASS [35] pre-training into ASR correction. Deliberation network [15, 16] applied error correction using both the first-pass transcripts as well as the input speech. However, all the end-to-end correction models are autoregressive and unsuitable for online deployment due to large latency, hindering the industrial application of ASR correction. Considering that there does not exit shuffle error in ASR correction, we propose a novel method to align the source sentences and target sentences towards global optimum based on edit distance, which can not only keep the matched tokens in alignment as many as possible, but also detect the substitution, deletion and insertion errors during alignment.

**Non-autoregressive Models**   NAR generation, which aims to speed up the inference of autoregressive model while achieving minimal accuracy drop, has been a popular research topic in recent years [13, 23, 30]. Gu et al. [8], Ma et al. [24], Shu et al. [34] approached this problem with a set of latent variables. Shao et al. [32], Ghazvininejad et al. [7], Li et al. [21] developed other alternative loss functions to help the model capture target-size sequential dependencies. Wang et al. [41], Stern et al. [38] proposed partially parallel decoding to output multiple tokens at each decoding step. Stern et al. [37], Gu et al. [9] proposed to generate target tokens in a tree-based manner, and used dynamic insertion/deletion to iteratively refine the generated sequences based on previous predictions. FELIX [25] performed NAR text edition by aligning source sentence with target sentence greedily. However, as shown in section 5.1, directly using existing non-autoregressive models such as Gu et al. [9] and Mallinson et al. [25] cannot get satisfying results on ASR error correction. In this paper, based on the characteristics of ASR recognized text, we develop FastCorrect, which builds edit alignment between the source sentences and target sentences to guide the error correction.

## 3 FastCorrect

FastCorrect leverages NAR generation with edit alignment to speed up the inference of the autoregressive correction model. In FastCorrect, we first calculate the edit distance between the recognized text (source sentence) and the ground-truth text (target sentence). By analyzing the insertion, deletion and substitution operation in the edit distance, we can obtain the number of target tokens that correspond to each source token after edition (i.e., 0 means deletion, 1 means unchanged or substitution, $\geq 2$ means insertion). FastCorrect adopts an NAR encoder-decoder structure with a length predictor to bridge the length mismatch between the encoder (source sentence) and decoder (target sentence). The obtained number of target tokens is used to train the length predictor to predict the length of each source token after correction, and to adjust each source token, where the adjusted source tokens are fed into the decoder for parallel generation. In the following subsections, we introduce the edit alignment, model structure and pre-training strategy in FastCorrect.

### 3.1 Edit Alignment

As shown in Figure 1, the edit alignment between each token in source and target sentences can be obtained through two steps: calculating the edit paths with minimum edit distance (the left and middle sub-figures), and choosing edit alignment with the highest n-gram frequency (the right sub-figure). In the next, we introduce each step in detail.

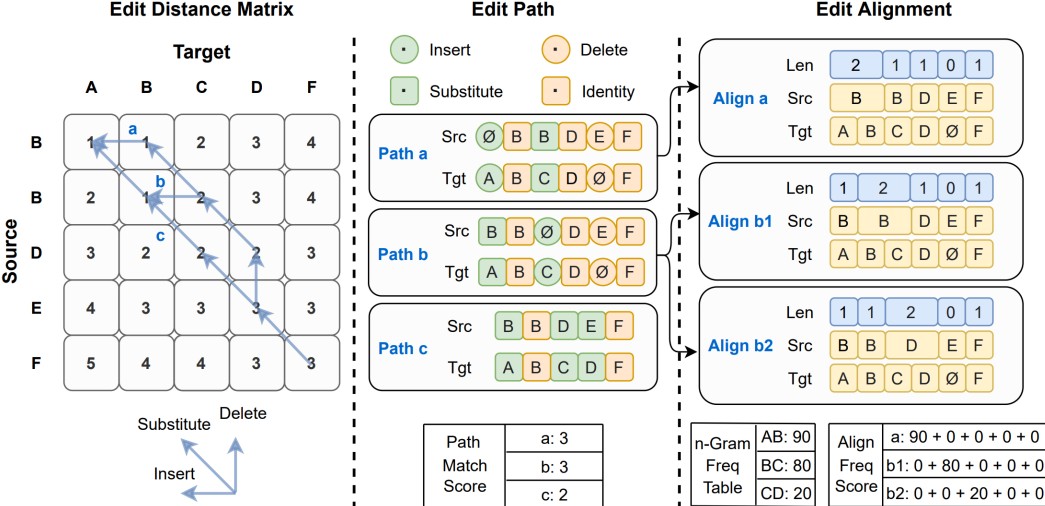

Figure 1: Illustration of the edit alignment between a source sentence "B B D E F" and a target sentence "A B C D F". The left part is the edit distance matrix calculated in a recursive manner. For example, the distance 2 in row D and column B means that the edit distance between source prefix sentence "B B D" and target prefix sentence "A B" is 2. The middle part shows all the possible edit paths with the minimum edit distance. Ø stands for an empty token. After filtering these paths with lower match scores, we can get all the possible token-level edit alignments, as shown in the right part. The alignment with the highest frequency score is selected as the final edit alignment to guide the error correction.

**Calculating Edit Path**    Edit distance measures the dissimilarity of two sentences by counting the minimum number of edit operations required to transform the source sentence into the target sentence[2]. The valid edit operations include token insertion, token deletion and token substitution.

Given source sentence $S = (s_1, s_2, ..., s_M)$ and target sentence $T = (t_1, t_2, ..., t_N)$, where $M$ and $N$ are the lengths of source and target sentences, the edit distance between $S$ and $T$ can be obtained by calculating the edit distance of prefix sentences recursively. The procedure is as follows:

$$D(i, j) = \min(D(i-1, j) + 1, D(i, j-1) + 1, D(i-1, j-1) + \mathbb{1}(s_i \neq t_j)). \qquad (1)$$

In the above equation, $D(i, j)$ is the edit distance of source prefix sentence $(s_1, s_2, ..., s_i)$ and target prefix sentence $(t_1, t_2, ..., t_j)$, $\mathbb{1}(\cdot)$ is the indicator function whose output is 1 when the condition is true otherwise 0. The boundary condition of $D(i, j)$ is $D(i, 0) = i, D(0, j) = j$.

The leftmost part of Figure 1 shows an example of the alignment between source sentence "B B D E F" and target sentence "A B C D F". By recursively calculating the edit distance, we can obtain all the possible edit paths that have the minimum edit distance. In this case, we have 3 possible paths with minimum edit distance 3, as path $a$, $b$ and $c$ shown in the middle part of this figure.

**Choosing Edit Alignment**    We introduce how to choose the edit alignment between each token in source sentence and target sentence based on the edit paths.

First, for the edit paths obtained from the above procedure, we calculate the match score of each path and only keep the paths with the highest match score. We define the match score of an edit path as the number of tokens that is not changed in this path. If an edit path has a higher match score, this path is more preferred since most tokens in the source sentence can be kept. As shown in the middle part of Figure 1, the match scores of the path $a$ and $b$ are both 3 because they both have 3 unchanged tokens: "B", "D" and "F", and the match score of path $c$ is only 2, which will be filtered[3].

---

[2]Standard edit distance usually transforms the target to the source. We change the order to transform the source to target to align with our scenario.

[3]Intuitively speaking, path $c$ does not make sense because it first changes token "D" to "C" and then changes "E" to "D". It is apparent that "D" should not be changed.

Second, we get the edit alignment set $E$ (where $e \in E$ represents a possible edit alignment for all tokens between the source and target sentences) from all the edit paths obtained by now by the following rules: 1) For a deletion, the source token is aligned with an empty target token Ø. 2) For a substitution or identity, we align the source token with the substituted token or unchanged token in the target sentence, respectively. 3) For an insertion, the target token has no source token to align with (e.g., token "C" in path $b$), and then the target token will be aligned with its corresponding left or right source token, resulting in different edit alignments (e.g., path $b$ can generate two alignments: $b1$ and $b2$, by aligning target token "C" to source token "B" (left) or "D" (right), respectively).

Third, we select the final edit alignment $e$ from the edit alignment set $E$ obtained in the second step, according to the frequency of n-gram tokens of the aligned target tokens. We first build an n-gram frequency table $G$ that contains the number of occurrences of every n-gram term in the training corpus, and then calculate the frequency score $Freq_{score}(e)$ for each edit alignment $e \in E$ on a source sentence $S$:

$$Freq_{score}(e) = \sum_{i=1}^{M} Freq(e[s_i]); \quad Freq(x) = \begin{cases} G[x], & len(x) > 1 \\ 0, & len(x) \leq 1 \end{cases}, \quad (2)$$

where $e[s_i]$ represents the target tokens aligned with the source token $s_i$ under alignment $e$, $M$ is the number of tokens in source sentence, $len(x)$ is the number of words in $x$ and $G[x]$ returns the frequency of $x$ in the n-gram table[4]. The frequency of all 1-gram is set to 0 because we are interested in differentiating token combinations. We choose the alignment $e \in E$ with the largest frequency score as the final edit alignment between each token in the source and target sentences. Doing so, we encourage the edit alignment that aligns the source token with more frequent n-gram target tokens. Taking the alignment $a$ in the rightmost part of Figure 1 as an example, where only the first source token "B" is aligned with more than 1 target tokens (i.e., "AB"). So the frequency score of alignment $a$ is equal to the frequency of "AB" in n-gram table $G$ (i.e., 90), which is larger than that of n-gram token "BC" in alignment $b1$ and "CD" in alignment $b2$. As a result, the alignment $a$ is selected as the final edit alignment and the length of target tokens aligned with each source token is "2 1 1 0 1", respectively.

## 3.2 Model Structure

We use Transformer [40] as the basic model architecture of FastCorrect, as shown in Figure 2. The encoder takes the source sentence as input and outputs a hidden sequence, which is 1) fed into a length predictor to predict the number of target tokens corresponding to each source token (i.e., the edit alignment obtained in the previous subsection), and 2) used by the decoder through encoder-decoder attention. The detailed architecture of length predictor is shown in the right sub-figure of Figure 2, which is optimized with MSE loss.

Thanks to the designs of the edit alignment and length predictor in FastCorrect, the deletion and insertion errors are detected by predicting a length of 0 or more than 1 on the corresponding source token through the length predictor. For substitution errors, the length predicted by the length predictor is 1, which is the same as the length of unchanged/correct source token. In this condition, the substitution error can be differentiated from the unchanged token by the decoder since it is different with the target token. These designs decrease the difficulty of error correction by using the length predictor to precisely detect the error patterns and using the decoder to focus on modification.

## 3.3 Pre-training

The high accuracy of ASR model, whose outputs are correct in a large proportion, makes the effective training cases for correction models limited since most words in a sentence are correct. To overcome this problem, we construct large-scale pseudo paired data to pre-train the FastCorrect model and then fine-tune on the original limited paired data. We crawl text data to construct a pseudo correction dataset by randomly deleting, inserting and substituting words in text data. To simulate the ASR error as close as possible to benefit the model training, we take two considerations: 1) The word is replaced with another word with similar pronunciation from a homophone dictionary when substituting, since substitution errors in ASR usually come from homophones. 2) The probability of modifying a word

---

[4]In the rightmost part of Figure 1, $E = \{a, b1, b2\}$, $b2[D]$ = CD, $G[CD]$=20.

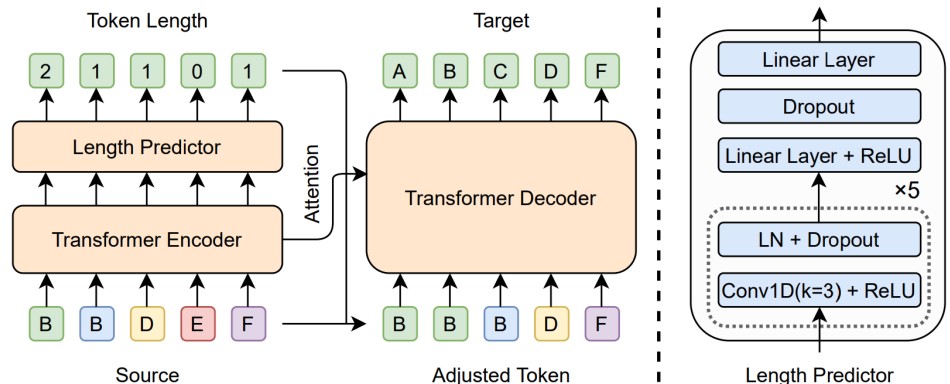

Figure 2: Model structure of FastCorrect.

is set to the word error rate of the ASR model. The probability distribution of deletion, insertion and substitution is set to the error distribution of the ASR model.

## 4 Experimental Setup

### 4.1 Datasets and ASR Models

We conduct experiments on two datasets, the public AISHELL-1 dataset and an internal product dataset.

**AISHELL-1** AISHELL-1 [3] is an open-source Mandarin speech corpus with 178 hours of training data[5]. We use the ESPnet [42] toolkit to train an ASR model on AISHELL-1 dataset. Several techniques such as Conformer architecture [10], SpecAugment [29] and speed perturbation are utilized to improve the performance of this ASR model, resulting in a state-of-the-art WER of 4.46 and 4.83 on the validation and test set of AISHELL-1. The ASR model is an encoder-attention-decoder model with a 12-layer conformer encoder, a 6-layer conformer decoder, a hidden size of 512 and a conformer kernel size of 15, which is trained with cross-entropy loss on decoder output and an auxiliary CTC loss on encoder output. We use 8 NVIDIA V100 GPUs for training and the batch size is 32 sentences per GPU. The output unit is Chinese character. The trained ASR models transcribe AISHELL-1 to generate the paired data for error correction.

**Internal Dataset** Our internal dataset is a Mandarin speech corpus with 75K hours of training data. Our ASR model on internal dataset is a hybrid model, where the acoustic model is a latency-controlled BLSTM [43] with 6 layers and 1024 hidden units in each layer, and the language model is a 5-gram model with 414 million n-grams trained with 436 billion tokens. The trained ASR models transcribe the internel dataset to generate the paired data for error correction.

**Pseudo Data for Pre-training** We use 400M crawled sentences to construct the pseudo paired data for pre-training. Each word in the original sentence is noised with a probability of $p$, which is set to the word error rate of the ASR model. For a word to be noised, the probability of substitution, deletion or insertion is estimated from the transcription results of ASR model, which is mentioned in the previous paragraph.

For all the text data in the above three datasets, we learn the subword using SentencePiece [19] with a dictionary size of 40K.

### 4.2 Model Configurations of FastCorrect and Baseline Systems

We use the default Transformer architecture in FastCorrect, which consists of a 6-layer encoder and a 6-layer decoder with hidden size $d_{model} = 512$ and feed-forward hidden size $d_{ff} = 1024$.

---

[5]https://openslr.org/33

Our length predictor consists of 5 layers of 1D convolutional network with ReLU activation and 2 linear layers to output a scalar, all of which have a hidden size of 512. Each convolutional layer is followed by layer normalization [2] and dropout. The kernel size of the convolutional network is 3. FastCorrect is implemented on Fairseq [28]. We describe the baseline systems used in our experiments for comparison: two NAR models, LevT [9] and FELIX [25], and a Transformer based autoregressive (AR) model.

**LevT**   We compare FastCorrect with another NAR model called Levenshtein Transformer (LevT) [9], which also predicts the insertion and deletion of a token in the decoder with multiple iterations. Different from LevT that implicitly learns the insertion and deletion to refine the target sentence, FastCorrect explicitly leverages edit distance to extract the edit alignment (insertion, deletion and substitution) between the tokens in the source and target sentences, which can be more efficient and accurate. We train LevT for error correction with the default hyper-parameters in Fairseq[6].

**FELIX**   FELIX is an NAR model for text edition [25]. Different from FastCorrect that utilizes edit distance for accurate alignment, the alignment algorithm is based on finding the matched tokens between the source and target sentences greedily, which 1) will be incorrect if the same token appears many times, 2) cannot perform substitution directly when editing text (i.e., need first deletion and then insertion). We implement FELIX based on the official code[7].

**AR Model**   For the autoregressive model, we follow the standard Transformer encoder-decoder model adopted in machine translation while keeping the parameter amount comparable with FastCorrect. We use the standard settings and hyper-parameters for AR model training in Fairseq[8].

### 4.3   Training and Inference

We train all correction models on 4 NVIDIA Tesla V100 GPUs, with a batch size of 12000 tokens. We follow the default parameters of Adam optimizer [18] and learning rate schedule in Vaswani et al. [40]. All the corrections models are first pre-trained on the pseudo data corpus for 30 epochs and then fine-tuned on the AISHELL-1 or the internal dataset for 20 epochs. To simulate the industrial scenario, we test the inference speed of the correction models in three conditions: 1) NVIDIA P40 GPU, 2) 4-core CPU and 3) single-core CPU, where the CPU is "Intel(R) Xeon(R) CPU E5-2690 v4 @ 2.60GHz". The test batch size is set to 1 sentence to match the online serving environment.

## 5   Results

In this section, we first introduce the accuracy and latency of FastCorrect, and then compare Fast-Correct with Transformer based autoregressive model, LevT and FELIX. Then we conduct ablation studies to verify the effectiveness of designs in FastCorrect, including edit alignment (length predictor) and pre-training. Finally, we conduct more analyses to compare FastCorrect with other methods.

### 5.1   Accuracy and Latency of FastCorrect

We first report the accuracy and latency of different error correction models on AISHELL-1 and the internal dataset in Table 1. We have several observations: 1) Autoregressive (AR) correction model can reduce the WER (measured by WERR) of ASR model by 15.53% and 8.50% on the test set of AISHELL-1 and internal dataset, respectively. 2) LevT, a typical NAR model from NMT, achieves minor WERR on AISHELL-1 and even leads to WER increase on the internal dataset. Meanwhile, LevT can only speed up the inference of the AR model by 2-3 times on GPU/CPU conditions. 3) FELIX only achieves 4.14% WERR on AISHELL-1 and 0.27% WERR on internal dataset, which is much worse compared with FastCorrect, although the inference speedup is similar. 4) Our proposed FastCorrect speeds up the inference of the AR model by 6-9 times on the two datasets on GPU/CPU conditions and achieves 8-14% WERR, nearly comparable with the AR correction model in accuracy. We further analyze the difference between FastCorrect, LevT and FELIX in Section 5.4. The above

---

[6]https://github.com/pytorch/fairseq/tree/master/examples/nonautoregressive_translation
[7]https://github.com/google-research/google-research/tree/master/felix
[8]https://github.com/pytorch/fairseq/tree/master/examples/translation

Table 1: The correction accuracy and inference latency of different correction models. We report the word error rate (WER), word error rate reduction (WERR) and latency of the autoregressive (AR) and non-autoregressive (NAR) models (FastCorrect, LevT and FELIX). "MIter" is a hyper-parameter in LevT controlling max decoding iteration. The actual iteration can be smaller than "MIter" due to early stopping.

| AISHELL-1 | Test Set | | Dev Set | | Latency (ms/sent) on Test Set | | |
|---|---|---|---|---|---|---|---|
| | WER | WERR | WER | WERR | GPU | CPU*4 | CPU |
| No correction | 4.83 | - | 4.46 | - | - | - | - |
| AR model | 4.08 | 15.53 | 3.80 | 14.80 | 149.5 (1×) | 248.9 (1×) | 531.3 (1×) |
| LevT (MIter=1) [9] | 4.73 | 2.07 | 4.37 | 2.02 | 54.0 (2.8×) | 82.7 (3.0×) | 158.1 (3.4×) |
| LevT (MIter=3) [9] | 4.74 | 1.86 | 4.38 | 1.79 | 60.5 (2.5×) | 83.9 (3.0×) | 161.6 (3.3×) |
| FELIX [25] | 4.63 | 4.14 | 4.26 | 4.48 | 23.8 (6.3×) | 41.7 (6.0×) | 85.7 (6.2×) |
| FastCorrect | **4.16** | **13.87** | **3.89** | **13.3** | **21.2** (7.1×) | **40.8** (6.1×) | **82.3** (6.5×) |

| Internal Dataset | Test Set | | Dev Set | | Latency (ms/sent) on Test Set | | |
|---|---|---|---|---|---|---|---|
| | WER | WERR | WER | WERR | GPU | CPU*4 | CPU |
| No correction | 11.17 | - | 11.24 | - | - | - | - |
| AR model | 10.22 | 8.50 | 10.31 | 8.27 | 191.5 (1×) | 336 (1×) | 657.7 (1×) |
| LevT (MIter=1) [9] | 11.26 | -0.80 | 11.35 | -0.98 | 60.5 (3.2×) | 102.6 (3.3×) | 196.5 (3.3×) |
| LevT (MIter=3) [9] | 11.45 | -2.50 | 11.56 | -2.85 | 75.6 (2.5×) | 118.9 (2.8×) | 248.0 (2.7×) |
| FELIX [25] | 11.14 | 0.27 | 11.21 | 0.27 | 25.9 (7.4×) | 43.0 (7.8×) | 90.9 (7.2×) |
| FastCorrect | **10.27** | **8.06** | **10.35** | **7.92** | **21.5** (8.9×) | **42.4** (7.9×) | **88.6** (7.4×) |

results demonstrate the effectiveness of FastCorrect in speeding up the inference of error correction while maintaining the correction accuracy.

## 5.2 Ablation Study

We conduct ablation study to verify the importance of the designs in FastCorrect, including the edit alignment (length predictor) and pre-training. For the setting without edit alignment, we train a predictor to predict the length difference between the input and the target, according to which the input is adjusted softly and fed into the decoder [12]. We show the results of each setting on the test set of both datasets in Table 2. We have several observations: 1) Removing edit alignment causes a large WER increase, which shows that the edit alignment is critical to guide the model to correct the error in the recognized text. 2) Pre-training is effective when the data

Table 2: Ablation study of each design in FastCorrect.

| Model | Internal Dataset | AISHELL-1 Dataset |
|---|---|---|
| No correction | 11.17 | 4.83 |
| AR model | 10.22 | 4.08 |
| - Pre-training | 10.26 | 16.01 |
| - Fine-tuning | 11.70 | 5.28 |
| FastCorrect | 10.27 | 4.16 |
| - Pre-training | 10.33 | 4.83 |
| - Fine-tuning | 11.74 | 5.19 |
| - Edit Alignment | 12.27 | 4.67 |

size in the fine-tuning stage is small. Removing pre-training causes a large WER increase on the AISHELL-1 dataset, which is much smaller than the internal dataset, especially in the AR setting. 3) Fine-tuning is necessary to ensure WER reduction in correction models, otherwise pre-training alone can result in worse WER than original ASR outputs. The above observations verify the effectiveness of each design in FastCorrect.

## 5.3 Comparison to AR Model with Shallow Decoder

As proposed in [14, 17], deep encoder and shallow decoder can reduce the latency of the AR model while maintaining the accuracy. We compare the accuracy and latency of FastCorrect to AR models with different combinations of encoder and decoder layers on the test set of both datasets in Table 3.

Table 3: Comparison of FastCorrect and the AR model with deep encoder and shallow decoder. "AR 8-4" means the AR model with an 8-layer encoder and a 4-layer decoder.

| Model | AISHELL-1 | | | Internal Dataset | | |
|---|---|---|---|---|---|---|
| | WER | Latency (ms/sent) | | WER | Latency (ms/sent) | |
| | % | GPU | CPU | % | GPU | CPU |
| No Correction | 4.83 | - | - | 11.17 | - | - |
| AR 6-6 | 4.08 | 149.5 (1×) | 531.3 (1×) | 10.26 | 190.6 (1×) | 648.3 (1×) |
| AR 8-4 | 4.14 | 120.5 (1.2×) | 427.6 (1.2×) | 10.28 | 144.1 (1.3×) | 542.0 (1.2×) |
| AR 10-2 | 4.23 | 84.0 (1.8×) | 317.6 (1.5×) | 10.33 | 100.8 (1.9×) | 431.2 (1.5×) |
| AR 11-1 | 4.30 | 66.5 (2.2×) | 281.0 (1.7×) | 10.44 | 79.1 (2.4×) | 372.3 (1.7×) |
| FastCorrect | 4.16 | **21.2** (7.1×) | **82.3** (6.5×) | 10.33 | **21.4** (8.9×) | **86.8** (7.5×) |

Table 4: Comparison of models on several metrics. We report: $P_{edit}$, among all the edited tokens, how many tokens actually need to be edited; $R_{edit}$, among all the error tokens, how many tokens are edited; $P_{right}$, among all the edited tokens, how many tokens are edited to target (i.e., error is corrected). $P_{edit}$ and $R_{edit}$ reflect the error-detection ability and $P_{right}$ reflects the error-correction ability. The word error rate reduction (WERR) of each model is also shown.

| Model | Internal Dataset | | | | AISHELL-1 | | | |
|---|---|---|---|---|---|---|---|---|
| | $P_{edit}$ | $R_{edit}$ | $P_{right}$ | WERR | $P_{edit}$ | $R_{edit}$ | $P_{right}$ | WERR |
| AR model | 78.8 | 17.5 | 50.5 | 8.50 | 83.9 | 33.6 | 54.0 | 15.53 |
| LevT | 57.9 | 16.7 | 33.6 | -0.80 | 71.7 | 14.4 | 38.4 | 2.07 |
| FELIX | 66.3 | 6.5 | 35.3 | 0.27 | 76.7 | 18.3 | 40.6 | 4.14 |
| FastCorrect | **79.0** | **18.4** | **47.6** | **8.06** | **83.7** | **34.5** | **50.1** | **13.87** |

FastCorrect achieves a much larger speedup than the AR models with a deep encoder and shallow decoder while maintaining similar accuracy with the AR models. To be more specific, FastCorrect has a better or comparable performance with auto-regressive model with 10 encoder layers and 2 decoder layers, while the speedup is about 5× higher.

## 5.4 Analysis of FastCorrect, LevT and FELIX

We perform an analysis on the ability of error detection and error correction between FastCorrect, LevT and FELIX to figure out why FastCorrect is better than LevT and FELIX.

If a source token is edited (including insertion, deletion or substitution) by a correction model, the edit result can be correct (same with the target token) or incorrect (different from the target token)[9]. Therefore, we can calculate several metrics to measure the advantage of a correction model: 1) $P_{edit}$, among all the edited tokens, how many tokens actually need to be edited. 2) $R_{edit}$, among all the error tokens, how many tokens are edited. 3) $P_{right}$, among all the edited tokens, how many tokens are edited to target (i.e., error is corrected). $P_{edit}$ and $R_{edit}$ measure the error-detection ability of correction models while $P_{right}$ measures the error-correction ability. The comparison of AR model, LevT, FELIX and FastCorrect on these metrics are shown in Table 4.

From Table 4, we can observe that compared with FastCorrect, the error-correction ability of both LevT and FELIX is inferior, since the ratio of tokens whose errors are corrected, $P_{right}$, of FastCorrect is higher than that of LevT or FELIX by a large margin. Moreover, the error-detection ability of LevT is weaker than FastCorrect, especially on the internal dataset, where the $P_{edit}$ of LevT is only 57.9%, meaning that 42.1% tokens edited by LevT are accurate already and these modifications only lead to WER increase. Thanks to the accurate edit alignment, FastCorrect can split the error detection and error correction into different modules, using length predictor to detect errors and decoder to

---

[9]For example, for a source sentence "ADC" and a target sentence "ABC", The edit result is correct if the source token "D" is edited to "B", while incorrect if edited to "E".

correct errors. Thus, FastCorrect can have comparable error-detection ability with AR model and high error-correction ability, enabling it to nearly match the accuracy of the AR model, and greatly outperform the two baselines.

## 6 Conclusions

In this paper, to reduce inference latency while maintaining the accuracy of conventional autoregressive error correction models for ASR, we proposed FastCorrect, a non-autoregressive model that leverages edit alignments (insertion, deletion, and substitution) between the tokens in the source and target sentences to guide error correction. FastCorrect utilizes a length predictor for error detection (predicting the number of target tokens aligned to each source token to detect insertion, deletion, and substitution/identity) and a decoder for error correction (modifying the error token into the correct one). Experiments on public AISHELL-1 and a large-scale internal dataset show that FastCorrect reduces inference latency of an autoregressive model by 6-9 times while maintaining its accuracy, and outperforms previous NAR models proposed for machine translation and text edition, which verifies the effectiveness of FastCorrect. Although FastCorrect is proposed for ASR, its design principle that first detecting errors and then correcting errors would be beneficial for general error correction. We will explore potential extensions of FastCorrect to other tasks and enable FastCorrect to leverage multiple ASR candidates [20].

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
