# OpenReview forum: "FastCorrect: Fast Error Correction with Edit Alignment for Automatic Speech Recognition"
_NeurIPS.cc/2021/Conference — NeurIPS 2021 Poster_

### Official Review · Reviewer_gLsB · 2021-07-10

**Rating:** 6
**Confidence:** 4

**Summary:**

This paper presents FastCorrect, a non-autoregressive text-editing model for ASR Error Correction. The authors propose a new text-editing architecture that is based on (1) sequence tagging model that predicts the final length of every source token in the target sentence and (2) a non-autoregressive decoder that predicts the target tokens in-parallel. To generate the training data, the source and the target sentences are aligned by calculating the edit distance and computing the edit path from the source to the target. Heuristics are applied to choose the path among all the paths with the minimal edit distance. The chosen path is used to create the labels for the length predictor and the non-autoregressive decoder.

The authors also perform data augmentation, generating data with synthetic errors using a homophone dictionary. This data is used to pre-train FastCorrect. Unfortunately it seems that this part is not reproducible, the crawled data is not published and it is not clear what homophone dictionary was used (I can only assume it is not open source).

The authors test FastCorrect on a public dataset and another internal dataset. On the public dataset there is a reduction of 0.67 WER. The authors then compare the WER and inference speed between FastCorrect and 3 other models: 2 publicly available non-autoregressive text-editing models (LevT and FELIX) and an autoregressive Transformer-based encoder-decoder model. FastCorrect shows superior performance in terms of WER compared to the other 2 non-autoregressive models and ~9 times faster inference than the autoregressive model.


**Ethical Concerns:**

None that I could think of.

**Limitations And Societal Impact:**

A agree with the authors that there is no obvious potential negative societal impact.

**Main Review:**

The paper is well written. ASR Error Correction is still an open area of research and new contributions are always welcome. The modeling idea is elegant and intuitive. The alignment algorithm is well presented and clearly explained (Figure 1 and the relevant description in section 3.1). It is encouraging to see that a text post-processing model can decrease WER.

The significance of the WER improvement is not clear to me. I am not sure whether 0.67 counts as a good improvement in this particular setup and whether the differences between the different error correction models are statistically significant. The evaluation is done on errors from a single ASR model, the results could be more convincing if they were evaluated using several (at least 2) ASR models. Given the fact that the  Error Correction model uses the same data as used to train the ASR model (​​AISHELL-1 training set), it is surprising to see that the ASR model could be corrected with a simple text-based model, even though the ASR model has acoustic information that is not available to the text-editing model. It could be really interesting to see a data analysis, or at least a case study, to understand what types of errors are actually being corrected.

While I understand that using an AR model with the same number of parameters is great to back up the claims about inference times, it could be interesting to see a WER of a state-of-the-art pre-trained sequence-to-sequence model (T5? BART?). If I would want to optimize WER I would start with a state-of-the-art pre-trained model, then if I would consider a NAR model for faster inference I would want to know the WER gap between them.

I did not read about LevT but from reading the FELIX paper, I think that claim #2 about FELIX is inaccurate, in Figure 2 in the paper (https://aclanthology.org/2020.findings-emnlp.111.pdf) it seems that there is a DEL-INS tag that allows substitutions. I also did not understand claim #1 and hoped the authors can provide a clarification. Why is the alignment incorrect if the same token appears many times? Maybe an example?

I find the analysis in section 5.4 problematic. It evaluates the error detection and correction on the sentence-level. Sentences can contain many errors, detecting or correcting part of them should count. For example if the sentence contained 10 errors, detecting/fixing 9 or 1 is not the same.

Section 3.1 lists different heuristics for choosing between the paths with the minimum edits (choosing the version that keeps most of the source tokens, using n-gram frequency table). While there is a clear and well explained intuition behind those heuristics, they complicate the algorithm and it is not clear whether they are actually necessary. An ablation study with a random shortest path could be helpful.

Reproducibility:

From the appendix it seems that the authors released only the recognition results of the ASR model and their evaluation scripts. It seems (from the description in the appendix, I did not look at the code) that the full training code (ASR and FastCorrect) was not released, so is the pre-trained or fine tuned version of FastCorrect. The authors claim that they will release those once the paper is accepted.
My main concern remains about the data used to pre-train FastCorrect. Even if the authors will release the data that they committed to release, the pre-training data is not published and it is not clear what homophone dictionary was used (probably not open source). This makes it impossible for future researchers to reproduce the results and build on this work.
This is crucial because in Table 2 we can see that without the pre-training there is actually no WER improvement on the public dataset using FastCorrect.

**Time Spent Reviewing:**

6

---

> ### Author Response · Authors · 2021-08-10
> **Response to Reviewer gLsB**
>
> Thanks for your comments on our paper. We reply to your questions as follows:
>
>
>
> **[Q1: About the reproducibility]**
>
>
>
> As the statement in supplementary materials, we will release pretrained FastCorrect model on 400M crawled data, which will include the pretraining 400M data itself and the homophone dictionary. We will release all items needed to reproduce our results on AISHELL-1 dataset.
>
> Our homophone dictionary is in url [1]. And the training code of FastCorrect is in url [2]. The pre-training data is too large, and thus we do not provide the url in the response period but will release it publicly together with our code. The public release of FastCorrect to GitHub is on-going and will contain all details including code, pretrain model/data, homophone dictionary, ASR model, etc.
>
>
>
> **[Q2: About the significance of WER improvement]**
>
>
>
> On AISHELL-1 dataset, according to the official result of Espnet, using Conformer architecture [3] results in 0.9 WER decrease on test set (6.7 -> 5.8, [4]). Adding SpecAug [5] into transformer model results in 0.6 WER decrease on test set (5.8 -> 5.2, [6]). FastCorrect reduces WER by 0.67 based on a strong SOTA baseline of 4.83 WER. The improvement is significant.
>
>
>
> Our internal dataset is an industry-scale dataset. According to common industrial practice, 5% WERR is significant enough to upgrade the online ASR model. FastCorrect achieves more than 8% WERR and has been deployed on our online serving, which is one of the largest ASR services in the world.
>
>
>
> The standard variance of accuracy of ASR models is within 0.02 WER and the improvement of FastCorrect over other error correction models are statistically significant (p<0.005).
>
>
>
> **[Q3: About results on different ASR models]**
>
>
>
> Besides encoder-attention-decoder (AED) model, we also performed experiments based on conformer-transducer ASR model [1] before paper submission which has an inferior accuracy to AED model but can better support streaming input, and thus more suitable for industrial deployment. We finally decided to report results on AED model because it could demonstrate that our method is able to improve accuracy of a SOTA ASR baseline. The correction results on AISHELL-1 based on conformer-transducer ASR model are as follows:
>
>
>
> |  Model   | Test Set WER  | Test Set WER Reduction  | Dev Set WER  | Dev Set WER Reduction   |
> |  ----  | ----  | ----  | ----  | ----  |
> | No Correction  | 5.05 | - | 4.57 | - |
> | AR Model  | 4.40 | 12.87 | 4.02 | 12.04
> | LevT (Miter=1) | 5.07 | -0.40 | 4.64 | -1.53
> | LevT (Miter=3)  | 5.10 | -0.99 | 4.67 | -2.19
> | FELIX  | 4.86 | 3.76 | 4.44 | 2.84 |
> | FastCorrect  | **4.51**  | **10.69** | **4.13** | **9.63** |
>
>
>
> The above results are consistent with AED results in Table 1 of our submission, showing that FastCorrect is nearly comparable with the AR correction model in accuracy and outperform baselines by a large margin. As Table 1 in our submission shows, FastCorrect speeds up AR model by up to 7 times and is faster than all baselines (We do not report latency above since it is almost the same as that in Table 1).
>
>
>
> **[Q4: About the types of errors being corrected]**
>
>
>
>
>
> On AISHELL-1 dataset, among all substitution errors that FastCorrect corrects, 50.98% are homophone errors (i.e., the error token and ground-truth token have the same pronunciation). FastCorrect can learn the error patterns of ASR model and correct errors based on these patterns (e.g., figuring out on which homophones that the ASR model is likely to fail and then correct those errors).
>
>
>
> **[Q5: About pre-training with T5 or BART]**
>
>
>
> In this version of FastCorrect, we have already used a large-scale (400M sentences, 35G text data) corpus to perform task-specific pretraining. As a comparison, the original BERT only used 16GB data to pretrain, BART used 160G, T5 used 745G.  We believe that the results on 400M pretraining can solidly reflect the effectiveness of FastCorrect. We will try T5/BART or using more data in pretraining in future work.
>
>
>
> **[Q6: About claims on FELIX]**
>
>
>
> **Claim #2**: In Figure 2 of FELIX paper, the DEL token and INS token is extracted, trained and predicted independently. In this example, three source tokens (“big very loud”) have tag “DEL” since they are unmatched in the target sentence -- “The noisy large cat”. We can see that two target tokens “noisy large” do not exist in the source sentence -- “The big very loud cat”. So a tag INS_2 is added to the token “loud” in “Mask” mode of FELIX. (In the “Infill” mode, a label INS is added to the token “loud” and INS label can be decoded to 1, 2, 3 or 4 words).
>
>
>
> Indeed, the alignment extracted from FELIX is many-to-many, e.g., aligning unmatched sequences “big very loud” with the unmatched sequences “noisy cat”. FELIX will add “DEL” token to all unmatched source tokens and add “INS” token to indicate how many target tokens need to be inserted.
>
>
>
> During inference, if the model performs a substitution, then the model needs to not only predict a DEL successfully, but also predict a “INS_1” (in Mask mode) or “INS” with 3 <PAD> token further predicted (in Infill mode), which is much harder than FastCorrect that using length 1 to stand for substitution or identity.
>
>
>
> In summary, the many-to-many alignment extracted from FELIX is not fine-grained enough and the substitution is hard to predict since the prediction is two-pass (DEL and INS are both needed to conduct a substitution). In contrast, FastCorrect can perform token-level alignment and directly perform substitution.
>
>
>
> **Claim #1**: We use an example to clarify claim #1.
>
>
>
> ASR result: *He eat end an apple pie here*
>
>
>
> Ground truth: *He eats an apple and an apple pie here*
>
>
>
> Note that there exist two “an” tokens and two “apple” tokens in ground truth. When the FELIX aligns the matched tokens in source sentence and target sentence, there will be 4 matched token pairs: “*He* - *He*”, “*an* - *an*(first)”, “*apple* - *apple*(first)”, “*Here* - *Here*”.
>
>
>
> However, since “*end*” is actually a mistake when recognizing “*and*”, the “*an apple*” in ASR result should be matched with the second “*an apple*”. With the help of n-gram score, FastCorrect is more likely to find correct alignment.
>
>
>
> **[Q7: About token-level error detection and correction ability]**
>
> Thanks for your comments! Actually, we have the token-level results before paper submission. To make the results concise and easy to understand, we only report sentence-level results in the submission.
>
>
>
> We report the token-level results here and will add them to the new version of the paper. The notation is the same as the submission. The only difference is that the notation is on token level. E.g., *P*$_{edit}$ means among all the edited tokens, how many tokens need to be edited. (The second to fifth columns are from internal dataset and the last 4 columns are from AISHELL-1.) These results are consistent with sentence-level results, showing that FastCorrect has an almost comparable error-detection and error-correction ability with AR model, which is much better than LevT and FELIX.
>
>
>
> | Model            | *P*$_{edit}$ | *R*$_{edit}$ | *P*$_{right}$ | WER | *P*$_{edit}$ | *R*$_{edit}$ | *P*$_{right}$ | WER |
> |  ----  | ----  | ----  | ----  | ----  | ----  | ----  | ----  | ----  |
> | AR model	| 78.83 |	 17.45	|  50.46	|    8.50	|	83.85	| 33.55	|  53.98 |     15.53
> | LevT	      | 57.88	| 16.66	|  33.60	|    -0.80	|	71.67	| 14.44	|  38.43  |     2.07
> | FELIX	      |  66.30	| 6.54	|  35.33	|    0.27	|	76.69	| 18.34	|  40.58   |    4.14
> | FastCorrect | **79.01**	| **18.38**	|  **47.62**	|    **8.06**	|	**83.72**	| **34.54**	|  **50.10**	|    **13.87**
>
>
>
> **[Q8: About ablation study with a random shortest path]**
>
>
>
> Randomly choosing a shortest path leads to a 0.5% WERR drop compared to FastCorrect with our path-selection algorithm that achieves 13.87% and 8.06% WERR on AISHELL-1 and internal datasets. Note that although we randomly select the shortest path, we are still performing token-level alignment and correction. Thus, the main effectiveness of FastCorrect is derived from the mechanism of performing token-level correction, which is achieved by our token alignment algorithm. Delicately choosing paths by match score and n-gram score will further boost accuracy (0.5% WERR) compared with randomly choosing. (See more about the importance of token-level correction mechanism in the last part of Q5 in the response to Reviewer GvdZ.)
>
>
>
>
>
> **Thank you again for your time and effort again on reviewing our paper! We will update the above discussions and results into the next version of our paper.**
>
>
>
> > [1] https://drive.google.com/file/d/1LUVrl1R5bWszM6rklkkcE6BtZloS82KE/view?usp=sharing
>
>
>
> > [2] https://drive.google.com/file/d/1Iu0jFN7tXrhiUgcRZqBnCrqaomWaQrvu/view?usp=sharing
>
>
>
> > [3] Anmol Gulati, James Qin, Chung-Cheng Chiu, Niki Parmar, Yu Zhang, Jiahui Yu, Wei Han, Shibo Wang, Zhengdong Zhang, Yonghui Wu, et al. Conformer: Convolution-augmented transformer for speech recognition. In INTERSPEECH, 2020.
>
>
>
> > [4] https://github.com/espnet/espnet/blob/master/egs/aishell/asr1/RESULTS.md#conformer-kernel-size--31-result
>
>
>
> > [5] Daniel S Park, William Chan, Yu Zhang, Chung-Cheng Chiu, Barret Zoph, Ekin D Cubuk, and Quoc V Le. Specaugment: A simple data augmentation method for automatic speech recognition. In INTERSPEECH, 2019.
>
>
>
> > [6] https://github.com/espnet/espnet/blob/master/egs/aishell/asr1/RESULTS.md#conformer-kernel-size--31--specaugment--lm-weight--00-result

---

> > ### Comment · Reviewer_gLsB · 2021-08-26
> > **Thank you for your responses**
> >
> > I would like to thank the authors for responding to my review comments and attaching the additional results. I have several followup questions.
> >
> >
> >
> > **Q1**
> >
> > How was the homophone dictionary created? How can a researcher build on top of your work in other languages (English?)?
> >
> > **Q2**
> >
> > “*The standard variance of accuracy of ASR models is within 0.02 WER*” - can you provide evidence for that claim?
> >
> >
> > **Q6 claim #2**
> >
> > I don’t agree with you claim "*DEL and INS are both needed to conduct a substitution*" - this is not accurate, the (single) tag that the model will predict is DEL-INS for substitution.
> >
> > FastCorrect is the one that needs 2 predictions in order to construct substitution, it needs to (1) predict DEL for a particular token and then (2) successfully predict 1 with the length predictor. While Felix just makes a single prediction of DEL-INS for the particular token.
> >
> >
> >
> > **Pre training section**
> >
> > You mention that “*the probability of modifying a word is set to the word error rate of the ASR model*” and “*the probability distribution of deletion, insertion and substitution is set to the error distribution of the ASR model*”. On what dataset do you calculate this word error rates and the insertion/deletion/substitution distribution?
> >
> > **Ablation study without pre-training**
> >
> > I think it could also be interesting to check how other NAR models perform without pre-training… Is it possible that LevT or Felix can actually be better without the pre-training? FastCorrect seems to be not useful without the pre-training  (at least on the public data, which is the most relevant dataset for future work).
> > This is important to guide future researchers what model to choose for each scenario. For example a researcher may want to build on top of your work in English and he will not have a homophone dictionary.

---

> > > ### Author Response · Authors · 2021-08-28
> > > **Thank you for your comments**
> > >
> > > Thanks for your further comments on our paper. We reply to your questions as follows:
> > >
> > > **[Q9: About the homophone dictionary]**
> > >
> > > The homophone can be easily created by the following steps.
> > >
> > > First, we can collect text data (e.g., from Wikipedia) and get a vocabulary.
> > >
> > > Second, we can obtain the pronunciation (i.e., the phoneme sequence) of every word in the vocabulary by open-source grapheme-to-phoneme tools (e.g., *g2pM (https://github.com/kakaobrain/g2pM)* package for Mandarin or *g2p (https://github.com/Kyubyong/g2p)* package for English).
> > >
> > > Third, we can calculate the similarity between the phoneme sequences of any two words in the vocabulary. Two words with a similarity higher than threshold are regarded as homophones. In our implementation, the similarity is the negative of ratio of *edit distance of two phoneme sequences* and *average length of two phoneme sequences*.
> > >
> > > **[Q10: About the standard variance]**
> > >
> > > Due to the large consumption of time and computing resources for pre-training, we only conduct repeat experiments with 4 different random seeds on the setting without pre-training on internal dataset. The standard variances of WER of AR model, LevT (Miter=1), FELIX and FastCorrect are 0.013, 0.015,0.013, 0.010, respectively.
> > >
> > > Although we do not perform repeat experiments on pre-training setting, we believe that pre-training on large-scale dataset will make the model performance more stable.
> > >
> > > **[Q11: About claim #2 in Q6]**
> > >
> > > We will give more explanations about claim #2:
> > >
> > > **FELIX needs two tags**: We discuss the “Infill” mode of FELIX rather than the “mask” mode since “DEL-INS” tag can only appear in “Infill” mode. As mentioned in Q6 (https://openreview.net/forum?id=N3oi7URBakV&noteId=DCamtD6mys2), when several consecutive tokens are not matched to the target, each source token is tagged with “DEL” and **only** the last source token is additionally tagged with INS, resulting in a DEL-INS tag. (In Figure 2 of FELIX paper, three source tokens “big very loud” are aligned with two target tokens “noisy large”. The first two token “big” and “very” have tag “DEL” and **only** the last token “loud” has tag “DEL-INS”). During the inference, if we need to substitute the first source token “big” with the first target token “noise”, then the model needs to firstly predict a “DEL” tag for token “big” and secondly predict a “DEL-INS” token for token “loud”. In this condition, the prediction is two-pass.
> > >
> > > **FastCorrect needs only one prediction**: For FastCorrect, there is no “DEL” token. The deletion and insertion are decided only by the length predictor through outputting a length of 0 and larger than 1, respectively. For substitution, it is decided by the length predictor through outputting a length of 1. You might have concern that the identity (i.e., correct source token) also has a length of 1. However, from experiments we find that the decoder can successfully learn to differentiate the substitution (modifying token) and identity (keeping the token). In our preliminary study, we have tried to train length predictor with an additional tag differentiating the substitution and identity. The results show that this tag cannot improve the accuracy.
> > >
> > >
> > > **[Q12: About probability]**
> > >
> > > The probability of modifying a word (i.e., adding noise) is set to be the word error rate of ASR model on the corresponding development dataset of AISHELL-1 and internal dataset, respectively. Also, the distribution of noise type (insertion, deletion and substitution) is also set to be the error distribution of ASR model on the corresponding development dataset of AISHELL-1 and internal dataset, respectively.
> > >
> > > **[Q13: About ablation study without pre-training]**
> > >
> > > We list the results of 4 models (FastCorrect, LevT, FELIX and AR model) with or without pre-training in the following table. The results are on the test set of internal dataset.
> > >
> > > |  Model   | WER w/o pre-training  | WER Reduction w/o pre-training    | WER w/ pre-training | WER Reduction  w/ pre-training |
> > > |  ----  | ----  | ----  | ----  | ----  |
> > > | No Correction  | 11.17 | - | 11.17 | - |
> > > | AR Model  | 10.26 | 8.15 | **10.22** | **8.50**
> > > | LevT (Miter=1) | 11.28 | -0.98 | **11.26** | **-0.80**
> > > | LevT (Miter=3)  | 11.49 | -2.86 | **11.45** | **-2.50**
> > > | FELIX  | 11.15 | 0.18 | **11.14** | **0.27** |
> > > | FastCorrect  | 10.33  | 7.52 | **10.27**  | **8.06** |
> > >
> > > The results show that when the training data is sufficient, FastCorrect outperforms LevT and FELIX without pre-training, and the effectiveness of pre-training is marginal (i.e., only 0.06 WER and 0.5% WERR for FastCorrect). So the key factor of accuracy improvement is FastCorrect itself rather than pre-training.
> > >
> > > The corresponding results on AISHELL-1 are as follows:
> > >
> > > |  Model   | WER w/o pre-training  | WER Reduction w/o pre-training    | WER w/ pre-training | WER Reduction  w/ pre-training |
> > > |  ----  | ----  | ----  | ----  | ----  |
> > > | No Correction  | 4.83 | - | 4.83 | - |
> > > | AR Model  | 16.01 | -231.47 | **4.08** | **15.53**
> > > | LevT (Miter=1) | 8.78 | -81.78 | **4.73** | **2.07**
> > > | LevT (Miter=3)  | 8.61 | -78.26 | **4.74** | **1.86**
> > > | FELIX  | 4.86 | -0.62 | **4.63** | **4.14** |
> > > | FastCorrect  | 4.83  | 0.00 | **4.16**  | **13.87** |
> > >
> > > From the table, LevT and FELIX also cannot reduce WER without pre-training. It is worth noting that the problem of limited training cases (discussion about this problem is in Section 3.3 of our submission), which is severe on AISHELL-1, prevents correction models from improving accuracy. And the proposed pre-training method can alleviate this problem.
> > >
> > > The proposed pre-training method is general (which can boost AR model, LevT, FELIX and FastCorrect) and easy to leverage since it only uses text data and homophone dictionary that is easy to create with open-source packages as mentioned in Q9.
> > >
> > >
> > > **We will update the above discussions and results into the next version of our paper. We hope that our response can address your problems. We are willing to have further rolling discussion in case some explanations are not clear enough.**

---

> > > > ### Comment · Reviewer_gLsB · 2021-08-29
> > > > **Thank you for your responses**
> > > >
> > > > I would like to thank the authors for responding to my additional review comments and attaching the additional results. I have a few final comment below and would like to retain my score.
> > > >
> > > >
> > > > **Regarding the pre-training**
> > > >
> > > > I cannot address the results on the internal dataset as I don’t know anything about it and it will not be published. Your main claim should be supported with the publicly available dataset, and any internal results are complimentary. On this dataset, the pre-training is a **crucial step**, and actually it can be an **additional significant contribution**. You actually show that with your pre-training all the NAR models start to be effective, and this is an interesting result. So if you would provide a **detailed and fully reproducible recipe for pre-training models for Error Correction** your paper could be stronger, but for this to happen I think you need to rewrite it and describe in much more detail how to construct the pre training corpus. Ideally (but optionally) if you could reproduce it for English like you described in the response and show some real examples of obtained homophones it could be even stronger.
> > > >
> > > >
> > > > **About claim #2 in Q6**
> > > >
> > > > While this is far from being the main point, I again need to kindly disagree with your interpretation. *During the inference, if we need to substitute the first source token “big” with the first target token “noise”, then the model needs to firstly predict a “DEL” tag for token “big” and secondly predict a “DEL-INS” token for token “loud”*. The model can also predict a DEL-INS tag for the token “big” and a DEL tag for “loud” (the inserter can still construct the target tokens successfully). In any case, this is **not a two-pass prediction**, as it is a tagging model and all the tags are predicted in a single pass… Unlike your model where you have tagging prediction + length prediction (so the claim *FastCorrect needs only one prediction* is inaccurate).

---

> > > > > ### Author Response · Authors · 2021-08-30
> > > > > **Thank you for your comments**
> > > > >
> > > > > **We sincerely appreciate for your time and efforts on reviewing our paper in detail!** Your comments are valuable, and our paper is strengthened with your help. We reply to your comments as follows:
> > > > >
> > > > > **[Q14: About pre-training corpus]**
> > > > >
> > > > > Following the steps in Q9 (https://openreview.net/forum?id=N3oi7URBakV&noteId=4jY5_MR_7xY), we download text data from (http://data.statmt.org/news-crawl/en/) and extract the homophones in English. The homophone dictionary is in (https://drive.google.com/file/d/1HCSGoUpIxZG-mjmHALPhwb_Iu7_D-3ag/view?usp=sharing). Here are some examples:
> > > > >
> > > > > The top-5 homophones of *"their"* are *"there them where then care"*.
> > > > >
> > > > > The top-5 homophones of *"would"* are *"wood woods could should good"*.
> > > > >
> > > > > The top-5 homophones of *"last"* are *"blast past least lost list"*.
> > > > >
> > > > > We can see that the homophones (or words with similar pronunciation) are found with our method.
> > > > >
> > > > > With homophone dictionary, constructing the pre-training corpus is not complicated. Every token in unpaired text data has a probability to be modified. The probability used in modification is explained in Q12 (https://openreview.net/forum?id=N3oi7URBakV&noteId=4jY5_MR_7xY).
> > > > >
> > > > > To be more specific, for a token *T*, when 1) performing deletion, we just remove token *T*; 2) performing substitution, we randomly use one of homophones of *T* from dictionary to substitute *T*; 3) performing insertion, we just randomly choose a new token *T’* from vocabulary mentioned in Q9 (https://openreview.net/forum?id=N3oi7URBakV&noteId=4jY5_MR_7xY) and insert *T’* before or after *T* randomly.
> > > > >
> > > > > We will update the above details about constructing homophone dictionary and pre-training corpus into the next version of our paper.
> > > > >
> > > > >
> > > > > **[Q15: About the contribution of pre-training]**
> > > > >
> > > > > When we conduct experiments on publicly available dataset, we find that pre-training is necessary for all correction models to be effective. Therefore, we introduce pre-training as a part of our method (in Section 3.3) and perform ablation study (in Section 5.2) on pre-training to highlight the effectiveness of pre-training.
> > > > >
> > > > > We agree with your opinion that pre-training is an additional contribution of our work. In the next version of our paper, we will point it out clearly in introduction section.
> > > > >
> > > > >
> > > > > **[Q16: About About claim #2 in Q6]**
> > > > >
> > > > > We apologize for using an inaccurate word *two-pass*. FELIX (as you mentioned) and FastCorrect (which has only length prediction and no tagging prediction) are both one-pass prediction. My point is that in FELIX, if a substitution is needed, then the model needs to predict **two** tags (a “DEL” tag for token “big” and secondly predict a “DEL-INS” token for token “loud”) simultaneously in one-pass prediction, which is harder than FastCorrect that only needs length predictor to predict a length of 1 (only **one** tag if we regard the output of length predictor as a kind of tag).

---

> > > > > > ### Comment · Reviewer_gLsB · 2021-08-30
> > > > > > **Short followup**
> > > > > >
> > > > > > Thank you again for this discussion.
> > > > > >
> > > > > > **Pre-training corpus**
> > > > > >
> > > > > > The homophone dictionary looks good. I agree that once you have the homophone dictionary the pre-training is not complicated. I do think that obtaining the homophone dictionary is not straight forward (I took a brief look at https://github.com/Kyubyong/g2p) and adding the exact steps (and code) for that could help. It should easy for future researcher to quickly obtain the homophone dictionary given a vocabulary. This is in line with my suggestion to frame it as additional contribution.
> > > > > >
> > > > > > **Regarding claim #2 for Felix**
> > > > > >
> > > > > > I agree that FastCorrect is also one pass as there is no tagging, only a length prediction. I still don't find it *easier* to predict the mentioned example. While Felix needs to successfully predict two tags, FastCorrect needs to successfully simultaneously predict correct lengths for "big" and "loud". It is true that FastCorrect has several options to predict relevant length, for example (1,1) or (0,2) or (2,0) - but so is the case for Felix that can predict (DEL, DEL-INS2) or (DEL-INS2, DEL) or (DEL-INS1, DEL-INS1) for masking, in case of infilling this could be (DEL, DEL-INS) or (DEL-INS, DEL). To summarize, both models need to make a consistent prediction **simultaneously** for all input tokens (in one pass), so I don't think one is easier from the other.

---

> > > > > > > ### Author Response · Authors · 2021-08-30
> > > > > > > **Codes for homophone dictionary**
> > > > > > >
> > > > > > > Thanks for your comments on our paper.
> > > > > > >
> > > > > > >
> > > > > > > The detailed steps are in Q9 (https://openreview.net/forum?id=N3oi7URBakV&noteId=4jY5_MR_7xY). Our codes of extracting homophone dictionary from raw txt data is in (https://drive.google.com/file/d/1VgloWikrr7IuiWVx1cDbojWeE9CulEx5/view?usp=sharing). These codes will also be included when we make FastCorrect open-source, which will help future researcher to quickly obtain the homophone dictionary given raw txt data.

---

### Official Review · Reviewer_CpNc · 2021-07-16

**Rating:** 5
**Confidence:** 5

**Summary:**

This paper addresses the topic of designing a non-autoregressive approach for correcting ASRs’ predictions. The motivation of applying a non-autoregressive approach is to allow parallel error corrections for the given hypothesis and reduce the inference latency. The proposed model has a length predictor which infer the target token length per input token, and use that info to map input tokens to target tokens and construct the final hypothesis. During training this target token length is calculated based on finding the minimal edit distance between source and target. In most cases there are multiple editing path with the same editing distance, and the work proposes to pick the one with highest alignment frequency between source and target. The proposed non-autoregressive approach significantly reduces the latency compared to autoregressive models, and provides lower WERs compared to existing non-autoregressive approaches.

**Limitations And Societal Impact:**

The paper needs to compare with more related approaches to inform whether the proposed work provides benefits over existing approaches.

**Main Review:**

The proposed approach infer per-token target length based on edit distance which is a nice design, and the empirical results showed significant improvement compared with other correction-based approaches. On the other hand, alternative to correcting the hypothesis with a non-autoregressive model, one can use the non-autoregressive model to rerank the hypothesis. The reranking approach does not need to infer the target length, and the training and inference pipelines are simpler. The paper lacks the comparisons with reranking approaches, and it’s unclear whether the proposed approach is more effective than reranking.

The framework is a minor extension of previous work, making its novelty limited.

**Time Spent Reviewing:**

4

---

> ### Author Response · Authors · 2021-08-10
> **Response to Reviewer CpNc**
>
> Thanks for your comments on our paper. We reply to your questions as follows:
>
>
>
> **[Q1: About the novelty]**
>
>
>
> To the best of our knowledge, there is no previous work designing end-to-end NAR model for ASR correction. We are the first to design NAR models for ASR correction that achieve comparable accuracy as autoregressive models. Furthermore, popular NAR models designed for machine translation don’t work well for error correction. Therefore, we design a novel NAR model based on edit alignment for ASR correction.
>
> In ASR correction, the correction accuracy will be hurt when modifying a correct token and thus precise information is needed to guide the correction. Since the ASR errors contain insertion/deletion/substitution which are monotonic (this is quite different from shuffle error which is popular in NMT), we calculate edit alignment between source and target, which is then used to train a length predictor to enable model to detect and correct errors in a fine-grained way. The mechanism of obtaining the token-level alignment between the source and target enables FastCorrect to correct the errors in ASR recognition results fast and accurately, which is non-trivial. So we respectfully disagree that FastCorrect is a minor extension of previous work.
>
>
>
>
>
> Moreover, experiments on public AISHELL-1 and a large-scale internal dataset show that FastCorrect reduces the inference latency of an autoregressive model by 6-9 times while maintaining the accuracy (8-14% WER reduction), and also outperforms NAR models designed for machine translation and text edition, which verifies the effectiveness of FastCorrect.
>
>
>
> **[Q2: About reranking result]**
>
>
>
> Indeed, we have already conducted reranking experiments before paper submission. Considering the reranking method cannot be fairly compared with FastCorrect (since reranking involves scoring on multiple candidates and choosing the best one, while FastCorrect corrects one candidate, the number of candidates they used are different), we do not include the reranking results.
>
>
>
> The WERR of LM reranking is 7.87% and 5.10% on AISHELL-1 and internal dataset, respectively, which is inferior to FastCorrect (i.e., 13.87% and 8.06%) by a significant margin. Moreover, LM reranking is slower than FastCorrect since it needs to score multiple candidates and FastCorrect only perform correction on one candidate.
>
>
>
> It is quite worth noting that reranking can be combined with FastCorrect to further achieve better correction accuracy since they improve ASR accuracy on multi-candidate or single-candidate level, respectively. Applying FastCorrect on reranking results can further improve the WERR to 14.90% and 8.86%, showing that these two methods are not conflict but complementary.
>
> **Thank you again for your time and effort again on reviewing our paper! We will update the above discussions and results into the next version of our paper.**

---

> ### Author Response · Authors · 2021-09-01
> **Willingness for Further Discussion**
>
>
>
> Thanks for your time on reviewing our paper.  By our previous author response, we sincerely hope that we can address your concerns by pointing out that 1) FastCorrect is the first end-to-end NAR model designed for ASR correction and outperforms popular NAR models designed for machine translation or text edition, 2) FastCorrect is better than reranking in terms of accuracy and latency, and more importantly, can be combined with reranking to further improve accuracy. Given that we still have time for rolling discussion, we wonder if there still exist some unclear explanations? Please let us know if you have questions and we are willing to clarify them.
>
> Best,
>
> Author

---

### Official Review · Reviewer_GvdZ · 2021-07-17

**Rating:** 6
**Confidence:** 4

**Summary:**

In this work the authors consider the problem of correcting the transcripts produced by the ASR system in the first pass. In previous works, this has been explored with attention-based encoder-decoder models which must process hypotheses sequentially left-to-right by feeding each output token. However, instead the authors propose to investigate non-autoregressive models where all outputs can be processed in parallel. This is achieved by first predicting the number of output characters corresponding to each source character and then simply modifying the input source sequence by repeating/deleting tokens as appropriate. The main benefit of the proposed approach is that it allows for significant savings in latency during inference.


**Limitations And Societal Impact:**

I don't see any negative societal impacts of this work.

**Main Review:**

Overall, the paper is generally well written and the main ideas are easy to follow. The central theme of the paper is relatively simple; I particularly liked the ideas around mapping the transcript to a single alignment sequence in a deterministic way to remove the inherent ambiguity in the definition of edit-alignments corresponding to the minimum edit distance. That being said, I have a few suggestions which I think the authors could address to strengthen the paper:

- **Related work**: I would also like to suggest that the authors include references to works which apply error correction using both the first-pass transcripts as well as the input speech in the so-called deliberation framework [Hu et al., 20] [Hu et al., 21]. I would also suggest that the authors reference this work [Guo et al., 19] which applies an error correction model in ASR.
K. Hu, R. Pang, T. N. Sainath and T. Strohman, "Transformer Based Deliberation for Two-Pass Speech Recognition," 2021 IEEE Spoken Language Technology Workshop (SLT), 2021.
K. Hu, T. N. Sainath, R. Pang and R. Prabhavalkar, "Deliberation Model Based Two-Pass End-To-End Speech Recognition," ICASSP 2020 - 2020 IEEE International Conference on Acoustics, Speech and Signal Processing (ICASSP), 2020.
J. Guo, T. N. Sainath and R. J. Weiss, "A Spelling Correction Model for End-to-end Speech Recognition," ICASSP 2019 - 2019 IEEE International Conference on Acoustics, Speech and Signal Processing (ICASSP), 2019, pp. 5651-5655, doi: 10.1109/ICASSP.2019.8683745.

- **Length prediction modeling and accuracy**: Something that the paper does not mention in sufficient detail is how the length prediction model is defined. Do you have a softmax over N possible values corresponding to 0 .. N-1? What size of N do you use? Please mention in the text. Also, what is the accuracy of the length prediction model on a held-out set?

- **Defining Latency**: The authors mention that one of the drawbacks of the AR-based model for spelling correction is that it has a very high latency and that it cannot be used for “online ASR tasks”. I think it is important to define these terms in the paper when they are introduced. For example, many works refer to the per-token latencies (which will still be extremely high in your setup since the error correction applies to the whole hypothesis). Also, if the encoders/decoders used LSTMs instead of transformers, then the sequence could in principle be encoded while the hypothesis was being decoded which would save some latency. I would suggest that the authors discuss more details in the paper.

- **Experimental Setup**: A number of details are missing in the experimental setup in Section 4. For example, what is the structure of the base ASR model? Is it a hybrid system or one based on encoder-decoder architectures? How was it trained? What output units does the model use, etc. Please clarify these details in the paper. Are the AI-shell models decoded without an LM?

- **Section 5.2**: Minor comment: The section which describes the ablation study is extremely brief, which makes it somewhat hard to fully understand. For example, the setup without edit alignment could be described in more detail, if possible.


**Time Spent Reviewing:**

5

---

> ### Author Response · Authors · 2021-08-10
> **Response to Reviewer GvdZ**
>
> Thanks for your comments on our paper. We reply to your questions as follows:
>
>
>
> **[Q1: About the references]**
>
>
>
> Thanks for your suggestions! We will cite the papers you mentioned in the next version of our paper.
>
>
>
>
>
> **[Q2: About the length prediction modeling and its accuracy]**
>
>
>
>  We model length prediction as a regression problem with L2-loss instead of a classification problem with cross-entropy loss. During inference, we round the prediction value to its nearest integer. We choose regression rather than classification because the length label is ordinal. For example, if the ground-truth label is 1.0, then predicting 2.0 is better than predicting 5.0 (although both of them are inaccurate and of the same error in classification), since 5.0 will generate 3 more wrong tokens. This can be better differentiated by regression than by classification.
>
>
>
> Here is the accuracy of length prediction. On AISHELL-1 test set, 98.94% token length is correctly predicted, 1.04% token length has a prediction error of 1.0 (i.e., the absolute gap between prediction length and ground truth length is 1), 0.02% token length has a prediction error of equal or greater than 2.0. On our internal test set, which is more challenging, the corresponding statistics are 95.02%/4.00%/0.98%. The MSE error of length prediction on AISHELL-1 and the internal test sets is 0.01 and 0.08, respectively. The high accuracy of length prediction ensures the effectiveness of our method for ASR correction.
>
>
>
>
>
> **[Q3: About defining latency]**
>
>
>
> In our paper, we use the sentence-level latency, i.e., the time to correct a whole sentence outputted by an ASR model. Many works on ASR error correction, including the reference [Guo et al., 19] mentioned in Q1, perform correction on the sentence level. Even so, the sentence-level correction can still be applied to the “online ASR tasks”. Specifically, the ASR model can output its recognized words one by one in a streaming manner. When a silence or the end of sentence is detected, the whole sentence is fed to the correction model and then the corrected sentence is returned to the user. This is a common practice in industry to incorporate the whole-sentence correction model into online ASR.
>
>
>
> Performing encoding while the ASR hypothesis is being decoded to further reduce the latency in FastCorrect is a good suggestion. We conduct experiments by constraining the encoder of FastCorrect to only see previous tokens. As a result, this constrained version of FastCorrect can take the decoded ASR hypothesis as streaming input and perform encoding while the ASR model is decoding.
>
>
>
> On our internal dataset, the result is 87.12 WER, which is much worse than the original ASR results (11.17 WER). We suspect that this huge accuracy drop is caused by the fact that encoder cannot see future tokens. So we use parameter *N* to control the number of future tokens that the encoder can see, i.e., when building the representation of current token, it needs to wait for *N* future tokens. When *N*=5, the correction model can achieve 10.77 WER (3.58% WER Reduction), which is largely inferior to the FastCorrect (i.e., *N*= ∞) which has an accuracy of 10.27 WER (8.06% WERR). When further increasing *N* to 10, the result is 10.33 WER (7.52% WERR), which is closer to FastCorrect. However, the latency reduction over FastCorrect is not significant (lower than 10% when *N*=10) due to the parallelism of hardware. How to reduce latency with streaming input is a good future direction. We will add discussions about this suggestion in the next version of our paper.
>
>
>
> **[Q4: About the ASR model]**
>
>
>
> For AISHELL-1, we train an encoder-attention-decoder model with a 12-layer conformer encoder, a 6-layer conformer decoder, a hidden size of 512 and a conformer kernel size of 15. This architecture is also adopted in [1][2]. The model is trained with cross-entropy loss on decoder output and an auxiliary CTC loss on encoder output. We use Adam optimizer with default hyper-parameters. We use 8 NVIDIA V100 GPUs for training and the batch size is 32 sentences per GPU. We also use a 5-gram language model. The output unit is Chinese character.
>
>
>
> For the internal dataset, as mentioned in paper, we use a hybrid model, where the acoustic model is a latency-controlled BLSTM with 6 layers and 1024 hidden units in each layer, and the language model is a 5-gram model. As for the training procedure, the hybrid model is trained firstly with cross-entropy loss and then MMI sequence training. The output unit is senone, which will be converted to Chinese character with HCLG graph during decoding.
>
>
>
> **[Q5: About the ablation study setup]**
>
>
>
> * For the setting without pre-training, we do not use 400M pseudo data to pre-train the correction model, whose results show that pre-training can improve the accuracy on internal dataset and (especially) on AISHELL-1.
>
>
>
> * For the setting without fine-tuning, we directly use model pre-trained on 400M pseudo data for ASR correction. The results show that the model with pre-training only cannot correct errors in ASR hypothesis (i,e, the WER after correction is higher than the original WER) and fine-tuning is necessary.
>
>
>
> * For the setting without edit alignment, we just predict the length of target sentence, instead of the length of each token in target sentence as FastCorrect does. Then we adjust the source sentence based on the predicted target sentence length and feed it into decoder. The detailed method for adjustment can be found in the end part of the first paragraph of Section 4.2 in [3]. The lower accuracy of this setting shows the effectiveness of the alignment module in FastCorrect.
>
>
>
> **Thank you again for your time and effort again on reviewing our paper! We will update the above discussions and results into the next version of our paper.**
>
>
>
> > [1] Zhang, Binbin, Di Wu, Zhuoyuan Yao, Xiong Wang, Fan Yu, Chao Yang, Liyong Guo, Yaguang Hu, Lei Xie, and Xin Lei. Unified streaming and non-streaming two-pass end-to-end model for speech recognition. arXiv preprint arXiv:2012.05481, 2020.
>
>
>
> > [2] Guo, Pengcheng, Florian Boyer, Xuankai Chang, Tomoki Hayashi, Yosuke Higuchi, Hirofumi Inaguma, Naoyuki Kamo et al. Recent developments on espnet toolkit boosted by conformer. In ICASSP 2021-2021 IEEE International Conference on Acoustics, Speech and Signal Processing, 2021.
>
>
> > [3] Junliang Guo, Xu Tan, Di He, Tao Qin, Linli Xu, and Tie-Yan Liu. Non-autoregressive neural machine translation with enhanced decoder input. In Proceedings of the AAAI Conference on Artificial Intelligence, 2019.

---

> > ### Comment · Reviewer_GvdZ · 2021-08-24
> > **Thanks for the responses**
> >
> > I would like to thank the authors for responding to my review comments and for running and reporting results on the "online" version of the model with limited label look-ahead. I would like to retain my scores as before.

---

### Author Response · Authors · 2021-08-24
**Willingness for Further Discussion**

Dear reviewers:

We sincerely appreciate all your high-quality comments on our paper. In the author response, we tried our best to resolve all the problems or concerns, which also strengthened our paper. Given that we still have time for rolling discussion, we wonder if there still exist some unclear explanations? Please let us know if you have further questions and we are willing to further clarify them.

Best,

Authors

---

### Decision · Program_Chairs · 2021-09-27

**Decision:**

Accept (Poster)

**Comment:**

Overview: The paper proposes a model that corrects ASR errors in a post-processing step. As reviewer GvdZ points out, there has been considerable recent work in this area. The model transforms the ASR output using a series of edit operations carried out using an encoder-decoder model which is augmented with duration (per input token). Empirical evaluations demonstrate show a small improvement over other comparable approaches.

Reviews: The author responses have addressed most of the concerns raised by two of the reviewers. The third reviewer felt the novelty of the proposed approach was incremental and limited. The meta review has taken into considerations the points raised by the three reviewers and the author responses.